# Pathway analysis in metabolomics: Recommendations for the use of over-representation analysis

Cecilia Wieder[1], Clément Frainay[2], Nathalie Poupin[2], Pablo Rodríguez-Mier[2], Florence Vinson[2], Juliette Cooke[2], Rachel PJ Lai[3], Jacob G. Bundy[4], Fabien Jourdan[2,5], Timothy Ebbels[1]*

1 Section of Bioinformatics, Division of Systems Medicine, Department of Metabolism, Digestion, and Reproduction, Faculty of Medicine, Imperial College London, London, United Kingdom, 2 Toxalim (Research Centre in Food Toxicology), Université de Toulouse, INRAE, ENVT, INP-Purpan, UPS, Toulouse, France, 3 Department of Infectious Disease, Faculty of Medicine, Imperial College London, London, United Kingdom, 4 Section of Biomolecular Medicine, Division of Systems Medicine, Department of Metabolism, Digestion, and Reproduction, Faculty of Medicine, Imperial College London, London, United Kingdom, 5 MetaToul-MetaboHUB, National Infrastructure of Metabolomics and Fluxomics, Toulouse, France

* t.ebbels@imperial.ac.uk

**Data Availability Statement:** The metabolomics and metadata reported in this paper are available via their respective MetaboLights or BioStudies

## Abstract

Over-representation analysis (ORA) is one of the commonest pathway analysis approaches used for the functional interpretation of metabolomics datasets. Despite the widespread use of ORA in metabolomics, the community lacks guidelines detailing its best-practice use. Many factors have a pronounced impact on the results, but to date their effects have received little systematic attention. Using five publicly available datasets, we demonstrated that changes in parameters such as the background set, differential metabolite selection methods, and pathway database used can result in profoundly different ORA results. The use of a non-assay-specific background set, for example, resulted in large numbers of false-positive pathways. Pathway database choice, evaluated using three of the most popular metabolic pathway databases (KEGG, Reactome, and BioCyc), led to vastly different results in both the number and function of significantly enriched pathways. Factors that are specific to metabolomics data, such as the reliability of compound identification and the chemical bias of different analytical platforms also impacted ORA results. Simulated metabolite misidentification rates as low as 4% resulted in both gain of false-positive pathways and loss of truly significant pathways across all datasets. Our results have several practical implications for ORA users, as well as those using alternative pathway analysis methods. We offer a set of recommendations for the use of ORA in metabolomics, alongside a set of minimal reporting guidelines, as a first step towards the standardisation of pathway analysis in metabolomics.

identifiers, or in the supplementary information of the relevant paper, detailed in Table 1 of the manuscript. The software developed in this study is available via a Jupyter notebook interface to enable reproduction of the simulations. The notebook, usage guidelines, dependencies, and processed metabolomics data are available via https://github.com/cwieder/metabolomics-ORA.

**Funding:** This research was funded in whole, or in part, by the Wellcome Trust [222837/Z/21/Z]. For the purpose of open access, the author has applied a CC BY public copyright licence to any Author Accepted Manuscript version arising from this submission. CW is supported by a Wellcome Trust PhD Studentship [222837/Z/21/Z]. RPJL receives support from the UK Medical Research Council (MR/R008922/1). JC is supported by a state-funded PhD contract (MESRI (Minister of Higher Education, Research and Innovation)). FJ is supported by the French Ministry of Research and National Research Agency as part of the French MetaboHUB, the national metabolomics and fluxomics infrastructure (Grant ANR-INBS-0010), and MetClassNet project (ANR-19-CE45-0021 and DFG: 431572533). TE gratefully acknowledges partial support from BBSRC grant BB/T007974/1, NIH grant R01 HL133932-01 and the NIHR Imperial Biomedical Research Centre (BRC). The funders had no role in study design, data collection and analysis, decision to publish, or preparation of the manuscript.

**Competing interests:** The authors have declared that no competing interests exist.

## Author summary

Metabolomics is a rapidly growing field of study involving the profiling of small molecules within an organism. It allows researchers to understand the effects of biological status (such as health or disease) on cellular biochemistry, and has wide-ranging applications, from biomarker discovery and personalised medicine in healthcare to crop protection and food security in agriculture. Pathway analysis helps to understand which biological pathways, representing collections of molecules performing a particular function, may be involved in response to a disease phenotype, or drug treatment, for example. Over-representation analysis (ORA) is perhaps the most common pathway analysis method used in the metabolomics community. However, ORA can give drastically different results depending on the input data and parameters used. Here, we have established the effects of these factors on ORA results using computational modifications applied to five real-world datasets. Based on our results, we offer the research community a set of best-practice recommendations applicable not only to ORA but also to other pathway analysis methods to help ensure the reliability and reproducibility of results.

## Introduction

Pathway analysis (PA) plays a vital role in the interpretation of high-dimensional molecular data. It is used to find associations between pathways, which represent collections of molecular entities sharing a biological function, and a phenotype of interest [1]. Based on existing knowledge of biological pathways, molecular entities such as genes, proteins, and metabolites can be mapped onto curated pathway sets, which aim to represent how these entities collectively function and interact in a biological context [2]. Originally developed for the interpretation of transcriptomic data, PA has now become a popular method for analysing metabolomics data [3,4]. There are several inherent differences between transcriptomic and untargeted metabolomics data, however, which must be considered when performing PA with metabolites. First, metabolomics datasets tend to cover a much lower proportion of the total metabolome than transcriptomic datasets do of the genome. Hence, metabolomics datasets tend to contain far fewer metabolites than transcripts found in transcriptomic datasets. Second, mapping compounds to pathways is not as straightforward as the equivalent mapping with genes and proteins, and there is often a significant level of uncertainty surrounding metabolite identification, both with respect to structures and database identifiers in any metabolomics dataset.

There are several methods for PA, which can be classed into three broad categories: over-representation analysis (ORA), functional class scoring (FCS), and topology-based methods [5]. In this paper, we focus on ORA, one of the most mature and widely used methods of PA both within the metabolomics [6,7] and transcriptomics [8] communities. ORA (referred to by some authors as metabolite enrichment analysis) has found widespread use in the identification of significantly impacted pathways in numerous metabolomics studies [9–13]. It works by identifying pathways or metabolite sets that have a higher overlap with a set of molecules of interest than expected by chance. The approach typically uses Fisher's exact test to examine the null hypothesis that there is no association between the compounds in the pathway and the outcome of interest [14].

To perform ORA, three essential inputs are required: a collection of pathways (or custom metabolite sets), a list of metabolites of interest, and a background or reference set of compounds. Pathway sets can be obtained from several databases, for example, the Kyoto Encyclopaedia of Genes and Genomes (KEGG) [15], Reactome [16], and BioCyc [17] databases, or

commercial counterparts such as the Ingenuity Pathway Analysis (IPA) database [18]. The list of metabolites of interest is generated by the user, most commonly obtained from experimental data and by using a statistical test to find metabolites whose levels are associated with an outcome (e.g., disease vs. control), and selecting a threshold (e.g., on the *p*-values) to filter the list. The background set contains all molecules which can be detected in the experiment. For example, in transcriptomic studies, this consists of all genes or transcripts which can be quantified. In targeted metabolomics, the background set would contain all compounds assayed; in untargeted metabolomics, all annotatable metabolites (i.e., all the features in a dataset that can be annotated to a compound name or ID). P-values for each pathway are calculated using a right-tailed Fisher's exact test based on the hypergeometric distribution. The probability of observing at least *k* metabolites of interest in a pathway by chance is given by (1):

$$P(X \geq k) = 1 - \sum_{i=0}^{k-1} \frac{\binom{M}{i}\binom{N-M}{n-i}}{\binom{N}{n}} \tag{1}$$

where *N* is the size of background set, *n* denotes the number of metabolites of interest, *M* is the number of metabolites in the background set mapping to the $i^{\text{th}}$ pathway, and *k* gives the number of metabolites of interest which map to the $i^{\text{th}}$ pathway. A visual representation of ORA is shown in Fig 1. Finally, multiple testing correction (to allow for the fact that, typically, the calculation is made for multiple pathways, rather than just one pathway) can be applied to obtain a final list of significantly enriched pathways (SEP).

Despite the widespread use of ORA in metabolomics [4] the community lacks a set of guidelines detailing its best use practices. Varying ORA inputs can result in large changes to outputs, which raises the question of how such parameters should be chosen in order to obtain the most reliable results. Moreover, as ORA was initially developed for use with transcriptomic data and later adapted for use on metabolomic data, there are certain considerations particularly important to metabolomics that may affect ORA results, such as the level of compound identification. Our aim here, therefore, is to investigate the robustness of ORA in typical metabolomics analysis, by examining the impact of varying the input data and parameters. The factors examined are: the background set, selection of differential metabolites, pathway database choice, organism-specific pathway sets, metabolite misidentification, and chemical bias of the assay. Using five experimental datasets, we vary the inputs, each time comparing to the original or standard settings, thus demonstrating the effect of these choices on the output lists of significant pathways. Based on our approach, we offer a set of recommendations for ORA applied to metabolomics data, as well as a set of minimal reporting recommendations which we hope can help contribute to future best-practice guidelines. It is hoped that this research will promote a deeper understanding of the use ORA in metabolomics, allowing researchers to better interpret their data in a pathway context.

## Results

### Nonspecific background sets result in erroneously high levels of enriched pathways

First, we examined several factors which are common to all ORA applications, beginning with the background set. Five publicly available metabolomics datasets have been used throughout this work (Table 1, see Methods). These datasets, obtained using untargeted mass-spectrometry (MS), were selected to encompass a diverse range of organisms, sample sources, and experimental conditions.

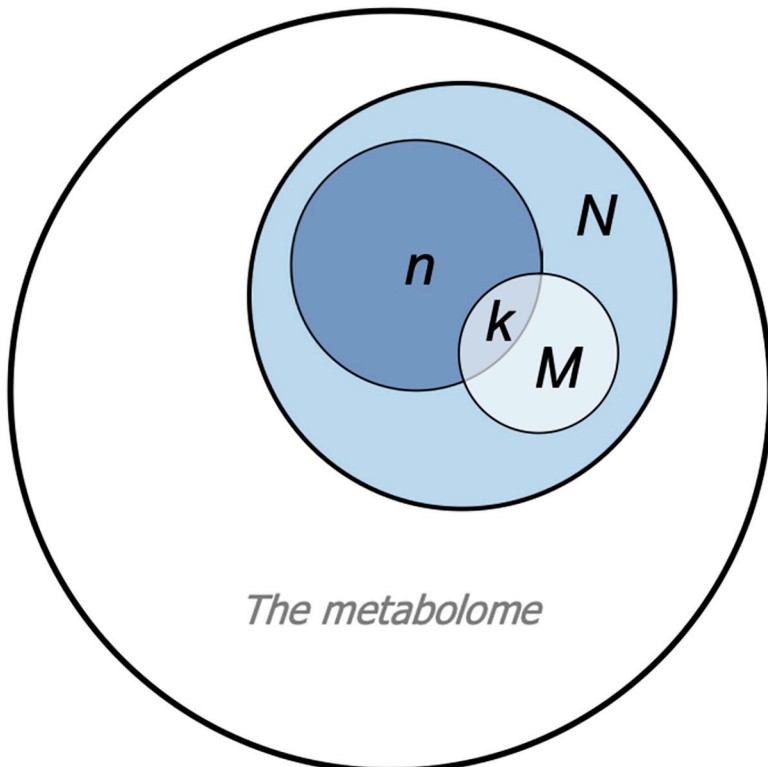

**Fig 1. Over Representation Analysis (ORA). Venn diagram representing ORA parameters corresponding to Eq 1.**
N represents compounds forming the background set, which covers part of the full metabolome. M represents
compounds in the pathway of interest. n represents compounds of interest (i.e., differentially abundant metabolites),
and k represents the overlap between the list of compounds of interest and compounds in the pathway.

The term background set (of size N, see Eq 1) is used to describe all the compounds identifi-
able using a particular assay. For example, for a targeted approach, this corresponds to the
compounds assayed; for an untargeted approach, this corresponds to all annotatable com-
pounds. Despite being a key parameter of ORA, specifying the background set is an often-

**Table 1. Summary of experimental datasets used in this work.** An asterisk (*) besides the MS platform indicates no chromatography/electrophoresis was used in the
assay.

| Author | Title | Organism | Analytical platform | Sample type | Total number of metabolites mapping to KEGG compounds | Study accession code/data availability |
|---|---|---|---|---|---|---|
| Labbé et al. | High-fat diet fuels prostate cancer progression by rewiring the metabolome and amplifying the MYC program | *Mus musculus* | UPLC-MS/MS | Tissue | 269 | MTBLS135 |
| Yachida et al. | Metagenomic and metabolomic analyses reveal distinct stage-specific phenotypes of the gut microbiota in colorectal cancer | *Homo sapiens* | CE-TOF MS | Stool | 286 | Supplementary table S13 of https://doi.org/10.1038/s41591-019-0458-7 |
| Stevens et al. | Serum metabolomic profiles associated with postmenopausal hormone use | *Homo sapiens* | UPLC-MS/MS | Serum | 362 | MTBLS136 |
| Quirós et al. | Multi-omics analysis identifies ATF4 as a key regulator of the mitochondrial stress response in mammals | *Homo sapiens* (HeLa cells) | Flow injection TOF MS* | HeLa cell | 1110 | Supplementary table S8 of https://doi.org/10.1083/jcb.201702058 |
| Fuhrer et al. | Genomewide landscape of gene-metabolome associations in Escherichia coli | *Escherichia coli* | Flow injection TOF MS* | E. coli | 2468 | S-BSST5 |

overlooked step. The use of a generic, non-assay-specific background set implies that non-observed compounds are considered in the Fisher's exact test formula, which, by definition, will always be absent from the list of metabolites of interest (of size n, Eq 1). We investigated the effect of using a nonspecific background set, consisting of all unique compounds present in the KEGG organism-specific pathway set, compared to an assay-specific background set, consisting only of compounds identified and present in the abundance matrix of each dataset. The nonspecific KEGG human background set contained considerably more compounds (3373) than any of the example datasets.

A clear discrepancy was observed in many of the pathway p-values when using the nonspecific vs. specific background set (Fig 2A). A greater proportion of pathways had lower p-values when using the nonspecific background set than the specific counterpart. Interestingly, some pathways were significant at $p \leq 0.1$ when using one background set but were not significant using the other, as evident in the upper right and lower left quadrants of Fig 2A. We also investigated the number of significantly enriched pathways (SEP) before and after multiple testing correction (using Benjamini-Hochberg False Discovery Rate (BH FDR)) when using the two different background sets (Fig 2B). When using the specific background set, there were far fewer SEPs at $p \leq 0.1$ (solid bars) and $q \leq 0.1$ (hatched bars) than there were using the nonspecific background set. Surprisingly, when using the specific background set (lighter coloured bars), two datasets contained no pathways which remained significant after multiple-testing correction (no hashed bars). Since our further analyses require several pathways to be enriched in the original datasets, we decided to use a significance threshold corresponding to an uncorrected p-value of $\leq 0.1$. While we do not recommend this threshold in practice as it is relatively liberal, this approach allowed us to demonstrate the characteristic behaviour of ORA across a wide range of datasets.

A key difference between the specific and nonspecific background sets used in the simulations in Fig 2 is the number of compounds they each contain. For the human datasets (Yachida, Stevens, and Quirós) for example, the nonspecific background set contained a total of 3373 unique compounds, whereas the specified background sets for these datasets ranged in size from 286 to 1110 compounds. It is therefore reasonable to ask whether the changes seen in Fig 2A and 2B could be due to the size of the background sets. Accordingly, we investigated how the size of the background set affects ORA results. In Fig 2C, we simulated a reduction in the number of compounds identified in the experiment and identify differentially abundant (DA) metabolites based on the compounds in the reduced background set. This could also reflect the differences in the number of metabolites identifiable using different platforms, for example, MS and NMR assays. In Fig 2D, we aimed to demonstrate how changing the number of compounds in the background set but keeping the number of DA metabolites static affects the number of SEP (hence changing the ratio of DA compounds to background set compounds). Both removal of compounds at random and non-DA compounds from the background set resulted in a decrease in the proportion of SEP ($p \leq 0.1$) as compared to using 100% of the compounds in the background set. Reduction of the background set at random (Fig 2C) resulted in a steady decrease in the number of significant pathways, as DA or non-DA compounds may be removed and the new list of DA metabolites is calculated based on the reduced background set. Reduction of the background set without removal of the original DA metabolites resulted in a much more variable decline in the number of significant pathways (Fig 2D). Datasets that had larger background sets to begin with, such as Fuhrer et al., appeared to be the least affected by the background set reduction. This is likely attributed to the fact that even when the reduced background set contained just 10% of the original compounds, it still contained over 240 metabolites. The trends observed in Fig 2D also imply that a

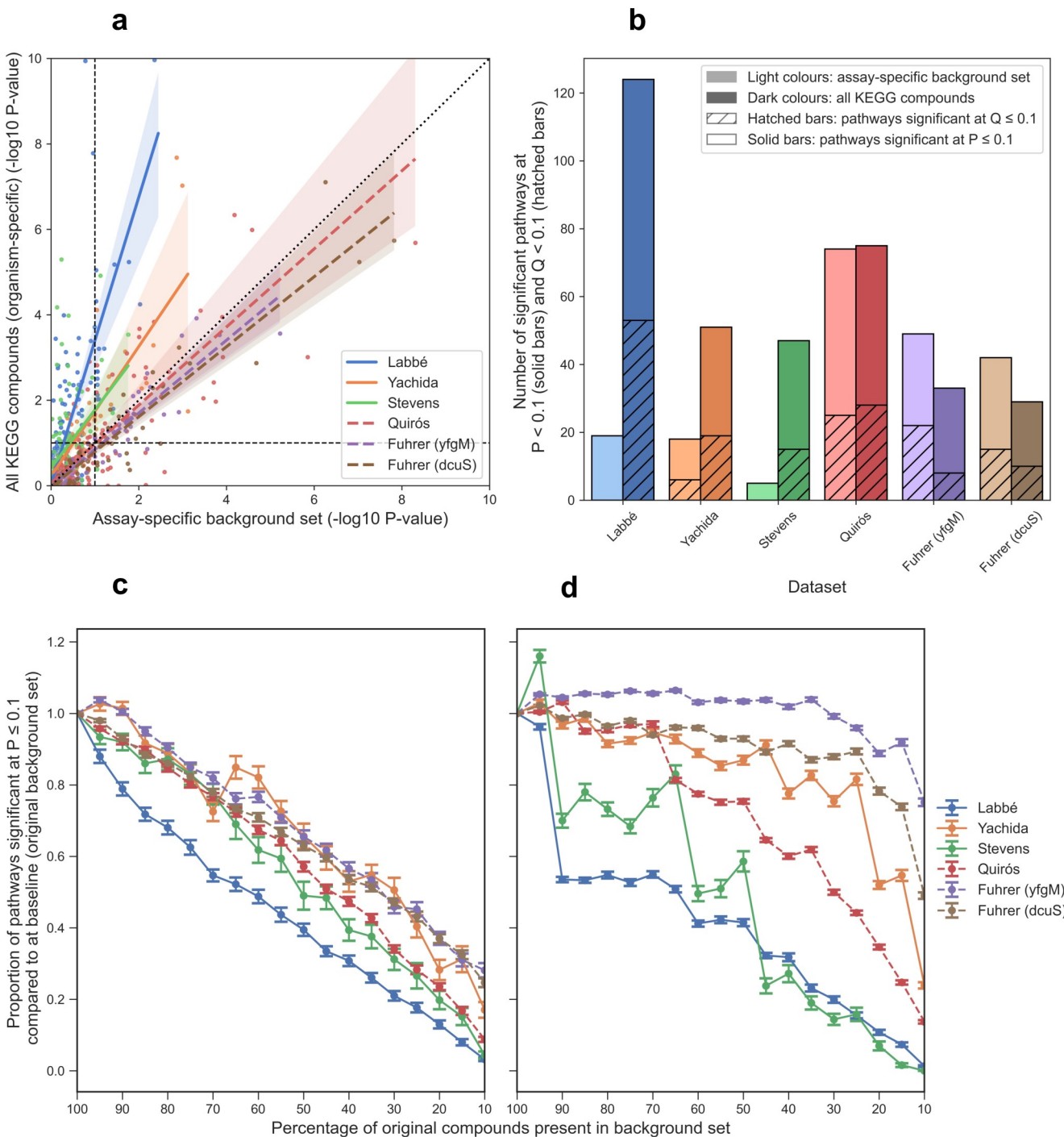

**Fig 2. Effect of background set. A)** Scatter plot of -log$_{10}$ p-values of pathways when using an assay-specific background set consisting of all measurable compounds in each dataset (x-axis) compared to using a non-specific background set containing all compounds mapping to at least one KEGG pathway (y-axis). Dashed black lines represent a p-value threshold equivalent to p = 0.1. Regression lines are shown with shading representing the 95% confidence interval. **B)** Number of pathways significant at p $\leq$ 0.1 (solid bars) and the number of pathways significant at q < 0.1 (hashed bars, BH FDR correction). Datasets are ordered by number of compounds mapping to KEGG pathways. **C and D)** The effect of reducing the size of the background set. **C)** Compounds were removed from the background set at random and DA metabolites were identified based on the modified background set. **D)** Only non-DA compounds were removed from the background set at random. In all panels a, c & d, dashed lines represent datasets where no chromatography/electrophoresis was used. Error bars represent standard error of the mean.

higher ratio of background set compounds to DA compounds provides more power for detecting SEPs.

## Increasing the number of differential metabolites can result in higher or lower numbers of significant pathways

The list of compounds of interest is a key parameter of ORA, as any compound falling below the significance threshold will not be able to contribute to the enrichment of a pathway. Methods used to select DA metabolites typically rely on p-values or q-values derived from a statistical test, for example when comparing metabolite abundances between study groups, or regression-based approaches for continuous outcomes. A threshold such as $q \leq 0.05$ is often used to select DA metabolites, however, as with all hypothesis testing this is an arbitrary choice. Furthermore, in untargeted metabolomics, hundreds or thousands of metabolites are often profiled and therefore multiple testing correction is essential. We investigated the effect of using varying significance levels and different multiple correction testing approaches to select metabolites of interest on ORA results. To this end, DA compound lists of increasing length were constructed by adding compounds, from lowest t-test *p*-value to highest, one at a time. T-tests were used to obtain the aforementioned p-values which reflect the significance of the difference in abundance of each metabolite between the two study groups. ORA was performed following the addition of each compound to the DA list. The number of SEPs detected using a DA list corresponding to Bonferroni adjusted *p*-values and BH FDR q-values at thresholds of 0.005, 0.05, and 0.1 was also determined. Note that here, we are discussing the significance level relating to selection of DA metabolites (the first step of ORA), not pathways (second step of ORA). Fig 3 shows an example of this procedure on the Labbé et al. dataset. Plots for all datasets are shown in Fig A in S1 Supporting Information. With the addition of each metabolite to the DA list, the number of SEPs tended to increase to a global maximum, followed by a decrease to zero where the DA list consisted of the entire background set. Several fluctuations can be observed as local minima and maxima in Fig 3, demonstrating that the addition of just a single compound can have a pronounced effect on the number of SEP. As expected, the list of DA metabolites determined by Bonferroni correction at varying alpha thresholds resulted in fewer significant pathways than using BH FDR correction. Generally, higher alpha thresholds resulted in more DA metabolites and hence more significant pathways. In the case of selecting metabolites based on BH FDR q-values however, more significant pathways were obtained using $\alpha \leq 0.05$ than $\alpha \leq 0.005$ or $\leq 0.1$. In summary, the addition of DA metabolites in order of significance will always result in an increase, followed by a decrease in the number of significant pathways. Thus, it is critical for practitioners to understand where their chosen significance threshold lies in this overarching trend.

## ORA results are influenced by pathway database choice, organism-specificity, and database updates

An important consideration when conducting any type of pathway analysis is the nature of the pathway sets used. Pathway sets can differ between databases in many ways, including the number of pathways present, the size of pathways, how pathways are curated (either manually or computationally, or a combination of both), pathway boundaries, and the organisms supported. We compared several properties of three pathway databases: KEGG, Reactome, and BioCyc. As this work focuses on metabolomics, only pathways which contain at least three metabolites were considered for the purposes of this paper, and genes and proteins were excluded from the pathway definition. Using human pathways as an example, as of December 2020, Reactome contained the highest number of pathways (1631), followed by HumanCyc

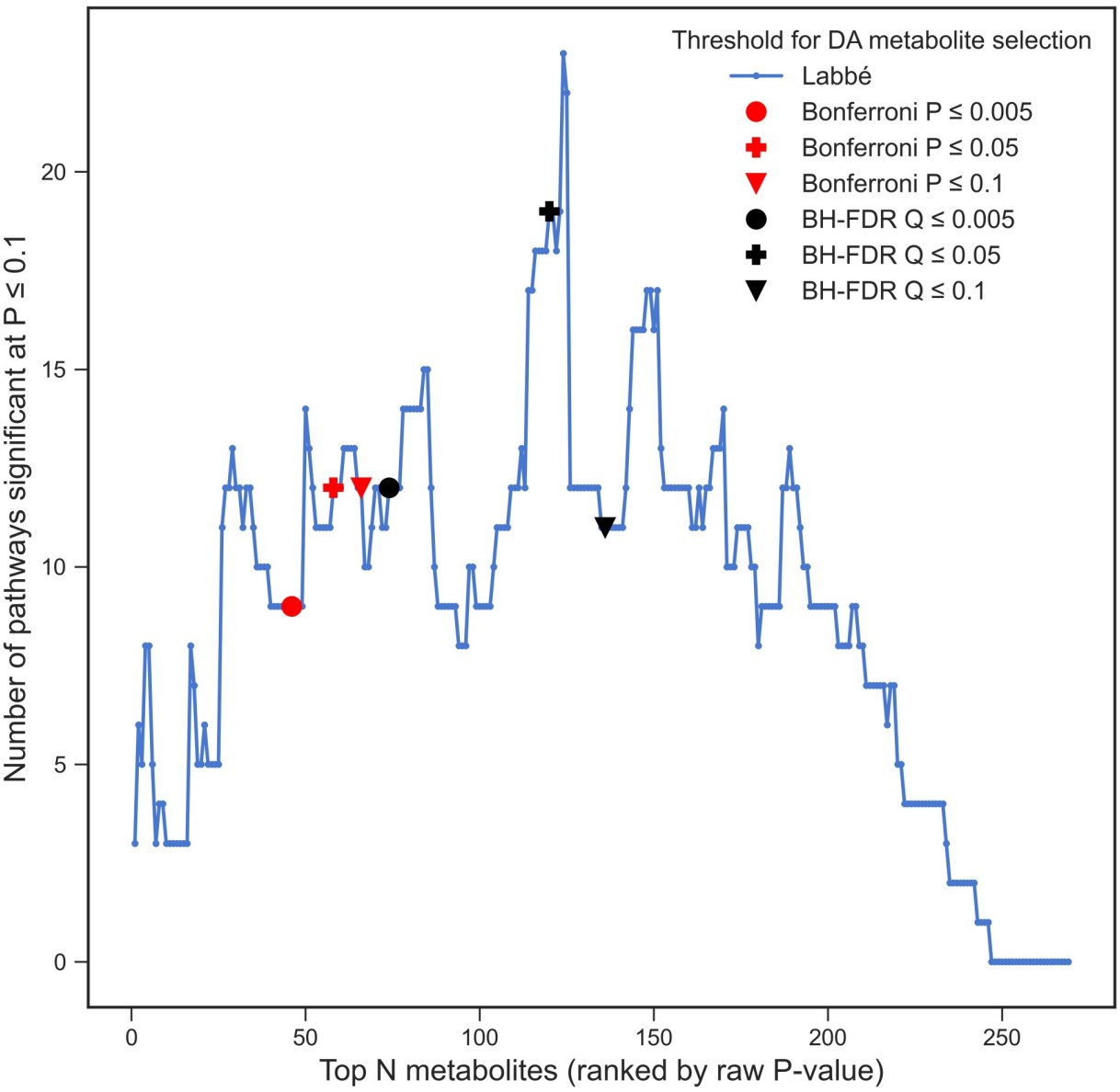

**Fig 3. Number of DA metabolites.** The effect of the number of DA metabolites in the list of metabolites of interest on the number of significant pathways (p ≤ 0.1) in the Labbé et al. dataset. Results corresponding to Bonferroni thresholds are denoted by red markers while those corresponding to BH FDR thresholds are denoted by black markers. Marker shape (circle, cross, or triangle) represents the adjusted p-value threshold for DA metabolite selection (0.005, 0.05, and 0.1 respectively).

(390) (part of the BioCyc collection) and KEGG, containing 261 pathways. A comparison of pathway sizes across the three databases can be seen in Fig 4A, in which HumanCyc pathways are the largest across the three databases, followed by KEGG and Reactome, based on median pathway size.

We next investigated the similarity of metabolite composition for KEGG and Reactome pathways. Identifiers for metabolites in each pathway were first converted to KEGG IDs and the ComPath [19] resource was used to find equivalent pathway mappings, linking KEGG and Reactome pathways with the same metabolic functions. We calculated the overlap coefficient (OC) for each of the 23 pairs of equivalent pathways. The OC (or Szymkiewicz–Simpson

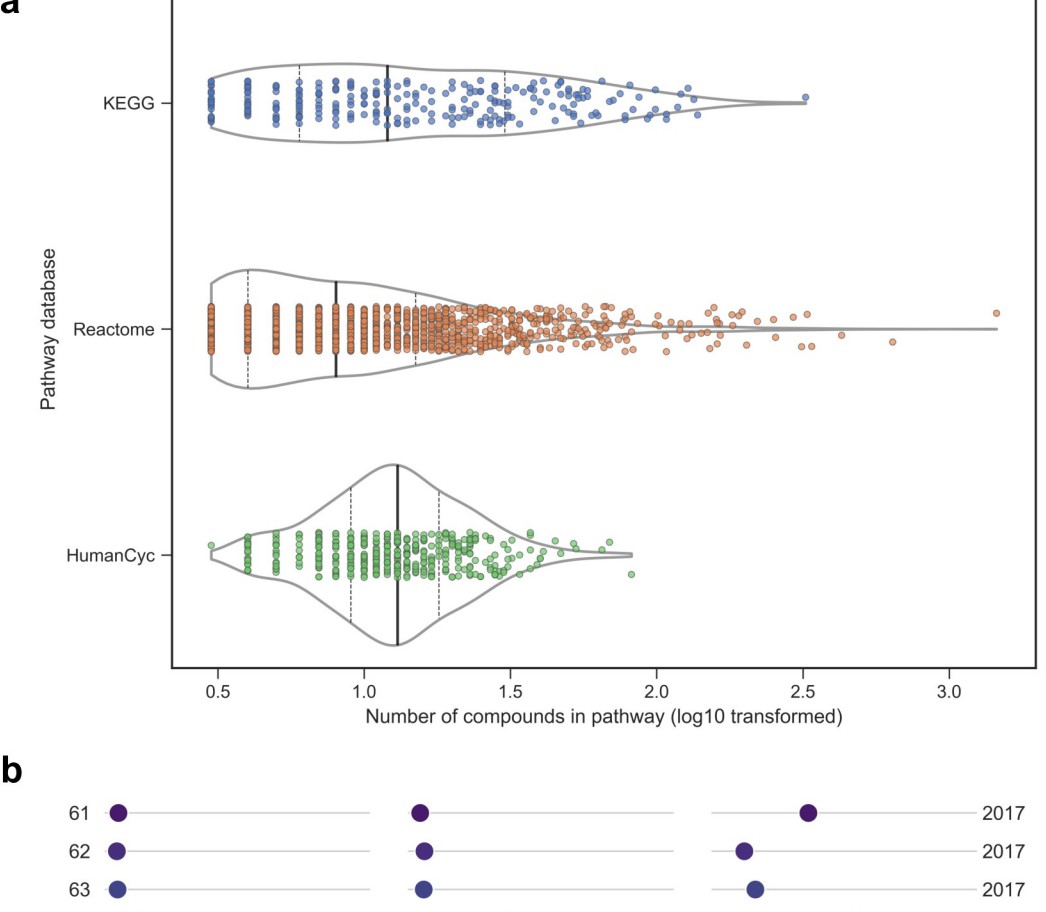

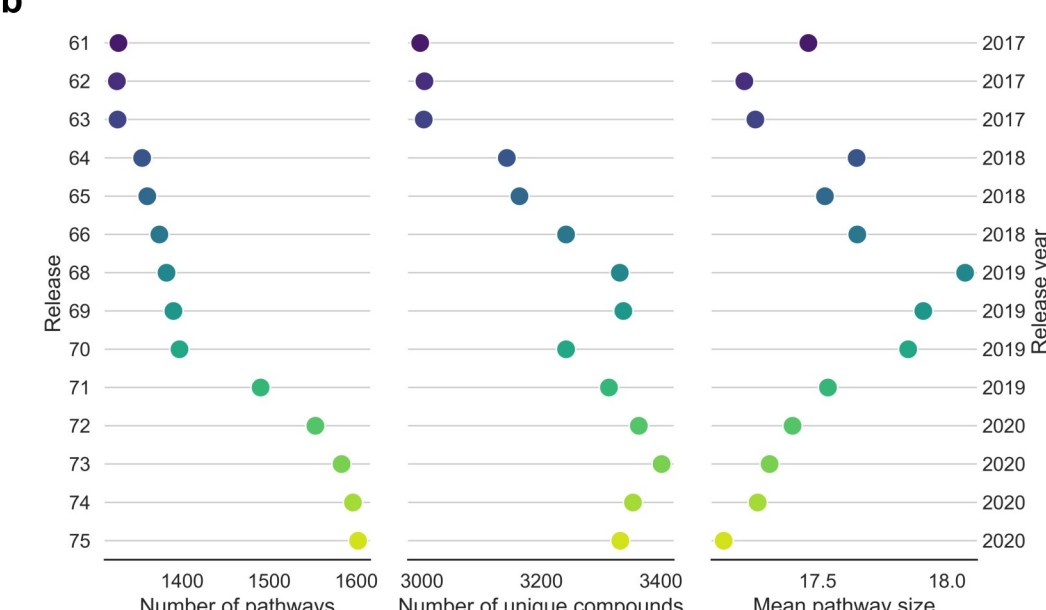

**Fig 4. Comparison of pathway databases and database updates. A)** Pathway size distribution of KEGG, Reactome, and HumanCyc databases. Violin plots show the distribution of pathway size (number of compounds, log10 transformed). Bold vertical lines show median, dashed vertical lines show lower and upper quartiles. **B)** Comparison of Reactome human pathway set (R-HSA) releases spanning the years 2017 (R61, June 2017) to 2020 (R75, December 2020). Data for release 67 was not available. Dot colour corresponds to release version, with lighter colours representing newer releases.

coefficient) compares two sets normalising by the size of the smallest set (see Methods). The OC may be more appropriate for comparison of metabolite sets than other similarity metrics such as the Jaccard index since it accounts for systematic differences in pathway sizes, which is the case here. The OC values were low (median = 0.33, interquartile range = 0.05–0.41), suggesting a low level of similarity in metabolite composition despite apparent equivalence of function. The same calculation was performed considering only genes in equivalent KEGG and Reactome pathways. 55 pathways were comparable, and while the OC values were larger than those derived from comparison of metabolite-only pathways (median = 0.64, interquartile range = 0.42–0.81), these also suggest moderate differences the gene composition of pathways from different databases.

To explore whether similar biological functions could be inferred from an ORA using different databases, we compared the SEPs obtained using the Yachida *et al.* dataset based on KEGG, Reactome, and HumanCyc pathways (Table A in S1 Supporting Information). By manual inspection of pathway names, there appeared to be low concordance between the results of the three databases in terms of biological function. Similar observations were also made in the other datasets. To quantify this effect, we pooled all metabolites from the significant pathways ($p \leq 0.1$) detected using KEGG and Reactome and calculated the OC between the two sets of compounds for each dataset. OC values ranged from 0.23 (Stevens dataset) to 0.62 (Labbé dataset) (Fig B in S1 Supporting Information), indicating low to medium consensus between ORA results derived using different pathway databases.

In addition to selecting a pathway database, many pathway databases offer both reference and organism-specific pathway sets. Reference pathway sets are not associated with any organism and can be useful when the organism under study does not have an associated pathway set. We compared basic properties of the KEGG human and KEGG reference pathways sets. The KEGG reference pathway set contained both more (377 vs. 261 pathways) and larger pathways (mean pathway size 45 vs. 30 compounds). The two pathway sets had a median OC of 0.92 (IQR = 0.83–0.97) for pathways with a common ID (e.g., Glycolysis: HSA00010/MAP00010), indicating a high level of similarity between the pairs but that analogous pathways are not identical. We performed ORA for each example dataset using both the organism-specific and reference pathway sets and compared the SEPs obtained (Table 2). While there was a large overlap, many more pathways were significantly enriched in the reference pathway set alone as opposed to in the organism-specific pathway set alone. This is likely due to the fact that the reference set contains more pathways, although not all of these may be of biological relevance to the organism in question.

A final consideration when selecting a pathway database is the version of the database one will use. Not all ORA tools will use the latest version of a certain pathway database available. The vast majority of pathway databases will undergo at least yearly updates, with some such as Reactome providing four major releases per year. To investigate how much impact pathway

**Table 2. Organism-specific vs. reference pathways.** Number of SEP (P ≤ 0.1) detected in both the KEGG organism-specific and KEGG reference pathway sets, and those significant in only one of the sets.

| Dataset | Common pathways | Organism-specific only | Reference only |
|---|---|---|---|
| Labbé | 19 | 0 | 6 |
| Yachida | 11 | 1 | 19 |
| Stevens | 5 | 0 | 1 |
| Quirós | 46 | 3 | 28 |
| Fuhrer (yfgm) | 27 | 0 | 26 |
| Fuhrer (dcus) | 27 | 0 | 23 |

database updates can have on ORA results, we obtained four years' worth of Reactome pathway sets spanning the period from June 2017 to December 2020. We compared three aspects of the Reactome human pathway sets (R-HSA) between each release: the number of pathways, the number of unique compounds in the database, and the mean pathway size (Fig 4B). As expected, the number of new pathways increased gradually from release to release, alongside the number of unique compounds. From 2017 to 2020, over 200 new pathways were added as well as almost 500 new compounds. Interestingly, the mean pathway size gradually increased from release 61 to release 68, after which it steadily decreased, but altogether remained between 17 and 19 compounds on average throughout the course of 14 releases.

## Metabolite misidentification results in both gain and loss of truly significant pathways

Next, we investigated some factors which are specific to metabolomics data, such as metabolite misidentification and assay chemical bias. A major bottleneck in untargeted metabolomics is the identification of compounds. In untargeted metabolomics, it is commonplace to putatively identify ("annotate") metabolites based on their physicochemical properties (e.g., m/z ratio, polarity) and similarity to compounds in spectral databases, and then confirm the identities of compounds of interest using chemical reference standards. Consequently, a large proportion of compounds in untargeted metabolomics assays are expected to have a degree of uncertainty in their identification, ranging from Metabolomics Standards Initiative (MSI) confidence levels 2–4 [20]. These levels refer to the minimum reporting criteria for metabolite identification proposed by the MSI, in which a level 1 identified compound is one that has been identified using an authentic chemical standard, as opposed to levels 2–4, which range from a compound putatively identified based on physicochemical and/or spectral similarities to compounds in a spectral library (level 2), to an unknown compound (level 4).

To compare the effects of metabolite misidentification on the number and identity of significant pathways detected using ORA, we introduce two new statistics: the pathway loss rate and the pathway gain rate (see Methods). The former describes how, as the data are degraded, some pathways are "lost" (no longer identified as significant) and others are "gained" (newly identified as significant). These are analogous to false-negative and false-positive rates, but account for the fact that we do not know the truly enriched pathways. For the purposes of this simulation, we make the assumption that all pathways significant at 0% misidentification are the "true" SEPs, and we compare these to the SEPs obtained at varying levels of simulated misidentification. The pathway loss rate refers to the proportion of SEPs present at 0% misidentification that are no longer present at $f$% misidentification, and the pathway gain rate refers to the number of SEPs not originally present at 0% misidentification which become significant at $f$% misidentification.

We simulated the effects of metabolite misidentification on ORA using KEGG pathways by replacing the true metabolites with false ones in two different ways: a) by similar molecular weight (20ppm window), and b) by identical chemical formula (see Methods). For both approaches, we calculated the pathway loss and gain rate for each dataset at 4% simulated misidentification which, although there are few published estimates of misidentification rates in metabolomics studies [21], endeavours to simulate a representative scenario (Fig 5). All the example datasets had nonzero pathway loss and gain rates at 4% simulated misidentification either by molecular weight or formula. Such findings suggest that even at a misidentification rate as low as 4%, it is likely that some pathways are significant simply as an effect of misidentification, and other pathways are not detected as significantly enriched due to the noise in the data caused by the misidentification. The similarity between the two modes of

misidentification may reflect the fact that most of the uncertainty in metabolite identification lies in associating a structure with a formula, rather than linking a formula to a mass. Pathway loss and gain rates from 1–5% misidentification are shown in Fig C in S1 Supporting Information. Pathway loss and gain rate results were similar for both misidentification by molecular weight and formula, likely owing to the fact that compounds with identical chemical formula share the same molecular weight.

### The chemical specificity of the assay influences the pathways discoverable using ORA

The analytical platform and specific assay used for a metabolomics study can be expected to introduce bias into the pathways which might be detected by ORA. Assays typically differ in their ability to detect compounds with different physico-chemical properties (e.g., polarity). While it is increasingly common for metabolomics experiments to incorporate multiple assays, most studies will still be biased in the compounds they can detect. We would expect to be able to access different pathways depending on the compounds assayed, resulting in disparate ORA results.

Using the Stevens et al. dataset as an example, which contains compounds identified using four different assay types, we mapped these compounds onto the KEGG pathway network using iPath 3.0 [22] (Fig 6A). It is evident that each of the four assay types covers a different area and proportion of the metabolic network. Even when the compounds from all four assays are taken together, large areas of the network remain unreachable, such as Glycan Biosynthesis and Metabolism, Lipid Metabolism, and Biosynthesis of Other Secondary Metabolites. It is therefore important to acknowledge this source of bias and recognise that certain areas of metabolism cannot be accessed. We further quantified this by computing the intersection between the pathways that were accessible using each assay type (Fig 6B). Indeed, the maximum number of pathways accessible using just one the assays (RP/UPLC-MS/MS with positive electrospray ionisation) was 63 (24.6%) out of a possible 256 KEGG human pathways containing at least two compounds. While there is a degree of overlap between pathways accessible using the different assays, a large proportion remains only accessible using a specific assay type.

### Discussion

As metabolomics continues to grow as a field of study with a multitude of applications within various disciplines, deriving meaningful conclusions from such data becomes increasingly important. ORA is one of the most popular approaches used to draw functional interpretations from metabolomics data. However, to date, there have been no published investigations of the consequences of varying input parameters on ORA results derived using metabolomics data. Understanding the sensitivity of ORA to tuning parameters, especially how it is influenced by metabolomics-specific factors, will play a crucial role in its successful application. In the present study, we sought to investigate the effects of varying inputs on ORA results, which we demonstrated using *in-silico* simulations based on five untargeted metabolomics datasets.

One of the most salient findings was the difference in the number of SEPs detected when using an assay-specific versus a nonspecific background set. The use of a nonspecific background set, such as all compounds present in the KEGG reference or human pathway set, for example, resulted in a drastic increase in the number of SEPs. In many ORA tools, use of a nonspecific background is typically the default option, and one that may lead users to believe that this is the 'correct' procedure. It is crucial however to understand that the consequence of not specifying a background set, which should contain all compounds that are realistically

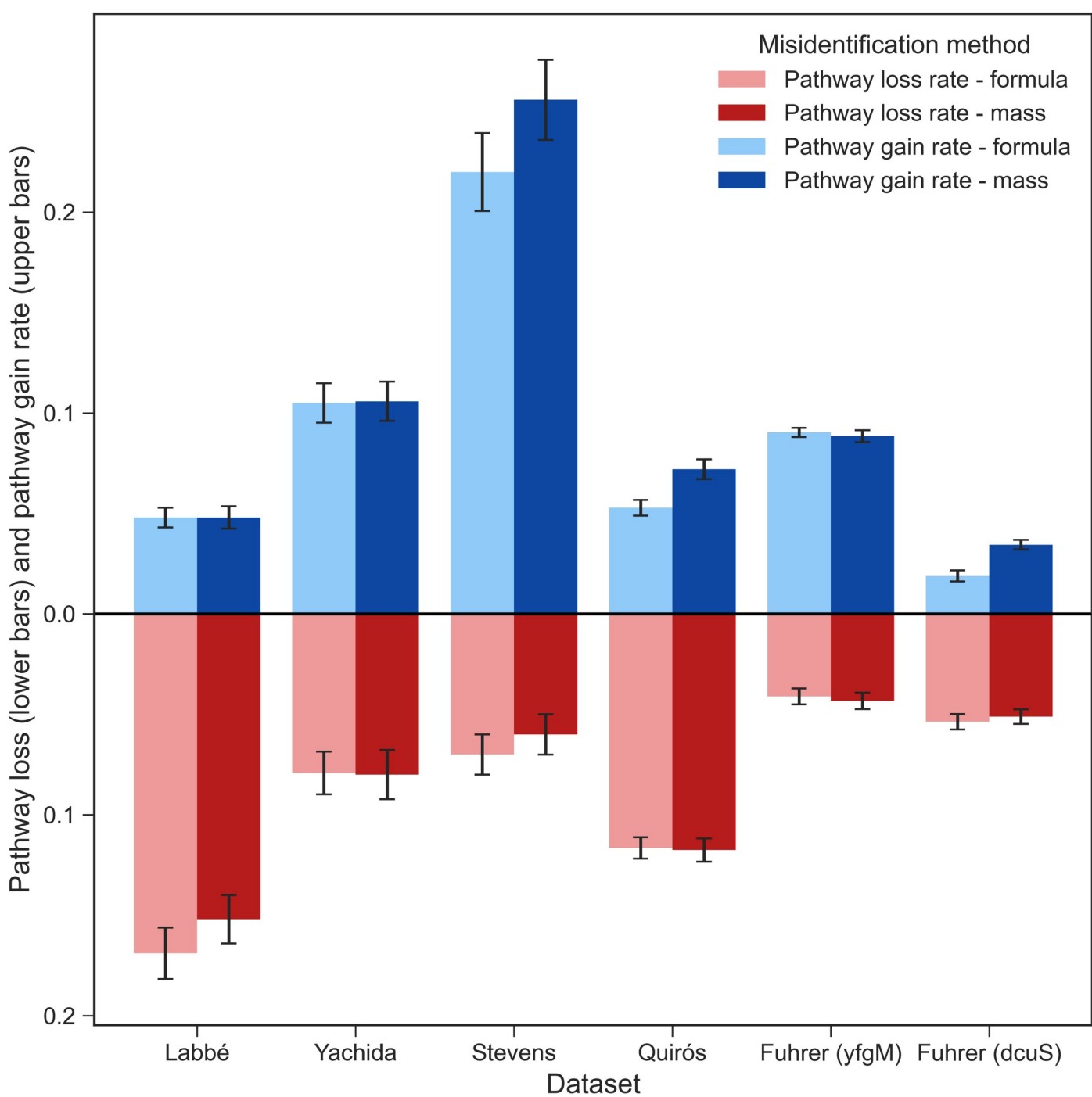

**Fig 5. Metabolite misidentification.** The effect of compound misidentification by molecular weight (20ppm window) (bars in dark colours) and chemical formula (bars in light colours) on the mean pathway loss rate (lower bars) and mean pathway gain rate (upper bars) averaged over 100 random resamplings at 4% misidentification. Error bars represent standard error of the mean.

observable, is that an assumption is being made that the compounds in the default background set are all equally likely to be detected in the experiment [24]. Such an assumption is highly unlikely to be true given that most technologies can only detect a small fraction of the metabolome and may lead to false-positive pathways. Additionally, the size of the background set is an

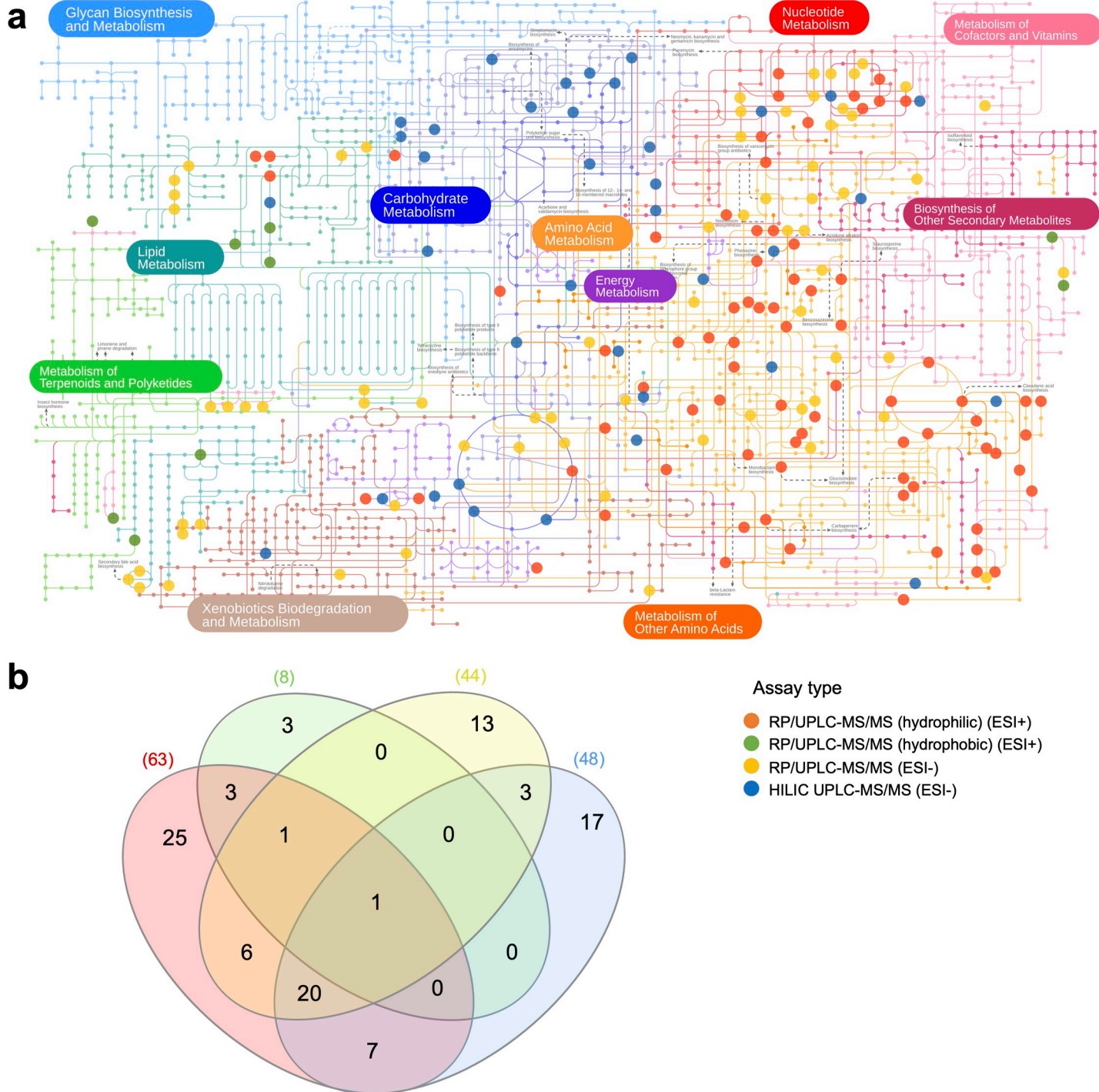

**Fig 6. The effect of assay chemical specificity on pathways accessible in the KEGG metabolic network.** Both figures a and b are based on the four assay types present in the Stevens et al. dataset. The colours in each subfigure correspond to the four assay types shown in the legend. **A)** KEGG reference metabolic network with compounds from each assay type highlighted on their respective pathways. KEGG network annotated using iPath 3 [22]. **B)** Venn diagram showing the number of KEGG pathways accessible using the compounds in each of the four assay types. Numbers outside the Venn diagram indicate the total number of pathways accessible with each assay type. Venn created using InteractiVenn [23].

important consideration, with larger sets generally yielding higher numbers of SEPs. MS-based approaches can usually detect a larger number of compounds than NMR-based methods, for example, at least for typical 1D NMR methods that are commonly used for profiling [25]. Users need to consider whether their metabolomics dataset is large enough to provide sufficient statistical power such that ORA results can be considered useful. Defining the ideal assay-specific background set for a particular dataset remains an area for further study. The approach used in this work was to use all identified compounds, which although conservative, is the safest approach minimising the number of false-positive pathways. The ideal assay-specific background set may be broader and is subject to considerations such as the compounds present in the spectral library used for identification, those above the detection limit and well quantified for the instrument used, and those expected to be present in the organism and sample source investigated.

The list of compounds of interest (often corresponding to metabolites differentially present between conditions in experiments) is an essential input for ORA and we have demonstrated that the way these compounds are selected greatly impacts PA results. It is important to select a threshold that strikes a balance between selecting too few compounds, therefore resulting in low power for the detection of significant pathways, or selecting compounds too liberally and losing power by introducing noise into the analysis. Visualisation of the curve of number of significant pathways vs. the number of compounds of interest (Fig 3) can be a useful way to determine the stability of the analysis to significance thresholds. Multiple testing correction should always be applied to all metabolite-level statistics before filtering them to produce the list of compounds of interest. We examined two of the most popular multiple testing correction methods: Bonferroni and BH FDR correction. By definition, Bonferroni correction tended to be more conservative, resulting in fewer compounds of interest, although this does not necessarily always correspond to fewer SEPs.

Unlike other fields (e.g., transcriptomics), the level of uncertainty surrounding compound identities remains a critical issue in metabolomics studies. While it is not possible to find a benchmark level of metabolite misidentification typically found in metabolomics studies, most studies will contain at least some misidentified compounds [26]. The level of misidentification will vary depending on the analytical platform used and remains a key bottleneck, more so in MS-based studies, where the number of metabolites detected often exceeds that of NMR-based studies [27]. In this study, we simulated metabolite misidentification by randomly swapping a small percentage of compounds in each of the datasets with compounds of either a similar molecular weight (± 20ppm) or an identical chemical formula. Even at a low level of misidentification of 4%, we found appreciable pathway loss and gain rates for all datasets. Hence, we suggest that ORA is sensitive to even low levels of metabolite misidentification, resulting in the emergence of false-positive and false-negative SEPs in the results.

Another essential input of ORA is the pathway database or list of metabolite sets used. The inherent differences between pathway databases will undoubtedly impact the PA results, regardless of the method used [28]. In the case of ORA, which is based on the hypergeometric formula, pathway size will influence results by rendering smaller pathways more significant and larger pathways less significant [29]. The number of pathways tested using ORA will also directly impact the adjusted significance level if multiple testing correction methods are applied, and the more pathways tested the more statistical power is lost. A related caveat is that the most widely used multiple testing approaches (e.g. Bonferroni, BH FDR) do not account for correlations between pathways and therefore such methods may be too conservative and undermine pathway significance [2].

A further important consideration for pathway database evaluation is the type of compound identifiers used in the pathway. KEGG and BioCyc use database-specific identifiers, whereas

Reactome uses ChEBI identifiers. It is necessary to convert the identifiers present in a metabolomics dataset to their database-specific equivalent, which often results in loss of information as not all identifiers will necessarily map directly to a database compound or be mapped to a pathway [30]. For example, in the Stevens et al. dataset, over 900 compounds were assigned to Metabolon identifiers, but less than half of these compounds could be mapped to KEGG identifiers. Another characteristic of metabolomics (and in particular lipidomics) is the discrepancy between the chemical precision of identification between the pathway databases and the dataset. For instance, in databases classes of lipids are often gathered into a single element (e.g., "a triglyceride") while lipidomics allows more in-depth annotation (e.g., "TG 16/18/18"). Computational solutions based on chemical ontologies exist to establish a link between dataset elements and pathway database ones [31], but this will also have an impact on PA results since several data elements will map to a single node in the pathway database.

The incompleteness of pathway databases, together with the evolution of pathway definitions between releases, are key factors highlighting the necessity of using an up-to-date resource; not doing so can have a detrimental effect on PA results [32]. Furthermore, the magnitude of changes across database releases demonstrated in this work suggests that ORA results are somewhat short-lived and perhaps valid only at a given time, hence they should be periodically revised using an updated database. Frainay et al. examined the coverage of analytes in the human metabolic network and found poor coverage of pathways involving eicosanoids, vitamins, heme, and bile acid metabolism [33]. Finally, although an extensive comparison of pathway databases is beyond the scope of this paper, several excellent studies have examined this in detail to which we refer the interested reader [28,34,35]. A general recommendation is to use multiple pathway databases and derive a consensus signature across these, if possible, reinforced by current knowledge of the underlying biochemistry of the system investigated. The use of integrative databases encompassing several pathway databases, such as the ConsensusPathDB [36], or interactive tools to simultaneously visualise pathways from different databases such as PathMe [37] may be beneficial and reflect ongoing efforts to harmonise pathway resources.

In this work we have focused on ORA, but many other PA methods exist [1,38,39]. While functional class scoring and topology-based methods can overcome certain limitations associated with ORA, such as the need to select compounds of interest, or not taking metabolite-level statistics into account, many of our findings are also relevant to these methods. Pathway database selection, metabolite misidentification rate, and assay chemical bias will impact the majority of metabolomics PA methods. Alongside the present work, further studies examining the input parameters of other PA methods for metabolomics data will be invaluable in establishing a set of best-practice guidelines for their application.

This study is limited by the lack of availability of a ground-truth dataset where the identities of enriched pathways are known. Possible sources of ground-truth data include simulations based on genome-scale metabolic models, in which enzymes in specific pathways are knocked out or the flux through reactions altered. Alternatively, one could insert artificial pathway signals into simulated or real data by altering the relative abundance levels of metabolites involved in the target pathways. Experimental datasets such as gene knockouts or knock-downs offer more realistic forms of ground truth datasets, which more accurately reflect the complexity of a biological system. Both simulated and experimental ground-truth datasets have limitations, however, such as the former being too simplistic, or the inability to pinpoint the exact pathway(s) affected by a perturbation in the latter. Nevertheless, such datasets might enable quantification of a wider variety of performance metrics than available here. Another limitation is that in the majority of examples, a p-value threshold of $P \leq 0.1$ was used without multiple testing correction to select SEPs. As metabolomics experiments usually identify far fewer compounds than transcriptomic experiments identify genes, ORA based on metabolites appears to have much lower power to

identify significant pathways and as such in the example datasets few, if any, pathways remained significant after multiple testing correction was applied.

The purpose of the present research was to evaluate the suitability of ORA for metabolomics PA and assess the effects of varying input data and parameters. We have investigated the three main input parameters: the background set, the list of compounds of interest, and the pathway database, as well as metabolomics-specific considerations such as metabolite misidentification and assay chemical bias. By means of *in-silico* simulations based on experimental datasets, all of the aforementioned variables have been shown to introduce varying levels of bias and uncertainty into ORA results, which has significant implications for those using ORA to analyse metabolomics data. In particular, use of an assay-specific background set is often ignored, yet has a critical effect on the output. Overall, this study has been the first detailed investigation into the application of ORA to metabolomics data, with wide-ranging findings that have implications not only to ORA but also a variety of other PA methods in metabolomics.

We therefore offer the community a set of recommendations for application, as well as suggested minimal reporting criteria, which may contribute to the future development of best-practice guidelines for the application of ORA to metabolomics data.

## Suggested recommendations for the application of ORA to metabolomics data

1. Specify a realistic background set based on the analytical platform used in the experiment. A conservative yet practical approach is to use all the metabolites that have been identified in the assay.

2. Use an organism-specific pathway set if the organism is supported by the pathway database.

3. Perform ORA using multiple pathway databases and derive a consensus pathway signature using the results if possible.

4. Use multiple-testing correction to select both DA metabolites and, where feasible, significant pathways.

## Suggested recommended minimal reporting criteria. Users should report

1. The statistical test/approach used for pathway analysis (e.g., Fisher's exact test)

2. The tool (and version) used to perform ORA.

3. The pathway database used, the corresponding compound identifier type (e.g., KEGG, ChEBI, BioCyc, etc.), its release number, and which organism-specific pathway set was used (if any).

4. Which compounds form the background set.

5. The multiple testing correction methods applied for i) selection of DA metabolites and ii) selection of SEP, alongside the adjusted p-value thresholds used.

## Methods

### Obtaining the list of metabolites of interest

**Summary of experimental datasets used.**    Five publicly available untargeted metabolomics datasets were used in this work (Table 1). The aim of this work was to select a small

sample of typical metabolomics studies to illustrate the effects of changing ORA parameters. The inclusion criteria for a dataset were: i) it should be publicly available, ii) it should contain over 100 annotated metabolites, and iii) there should be at least two study groups. For consistency, all datasets used in this work are based on mass-spectrometry (MS). The first dataset is available at MTBLS135 from the MetaboLights repository and consists of 12 Hi-Myc genotype and 12 wild-type *Mus musculus* tissue samples [40]. The second dataset from Yachida et al. 2019 [41] consists of 149 healthy control and 148 colorectal cancer human stool samples (stages I-IV). The third dataset is available at MTBLS136 and consists of 667 control samples and 332 estrogen users [42]. The fourth dataset is from Quirós et al. 2017 [43] from which we compared 8 HeLa cell replicates treated with actinonin to 8 HeLa cell replicates treated with doxycycline. The final dataset is available from EBI BioStudies (S-BSST5) and consists of >3,800 single-gene E. coli knockouts each with 3 biological replicates [44]. Data from the positive and negative ionisation modes was combined to provide the final matrix of putative compound identifications and relative abundances for each. We selected two knockout strains to investigate from this dataset which were amongst those with the highest effect size (based on the number of significant pathways detected using ORA): $\Delta$yfgM and $\Delta$dcuS. It is important to note that two datasets, Quirós et al. 2017 and Fuhrer et al. 2017, did not use any separation step in their analytical platform, and therefore there may be a higher degree of uncertainty in the metabolite identifications.

**Post-processing of metabolomics datasets.**   All metabolomics datasets and corresponding metadata used in this study are publicly available from the MetaboLights repository [45], the BioStudies database [46], or in the supplementary information of the original publication (Table 1). Details of metabolomics data pre-processing, as well as sample preparation, data acquisition, and compound identification can be found in the original publication for each dataset. For the purposes of this study, the pre-processed raw metabolite abundance matrices consisting of $n$ samples by $m$ metabolites were downloaded as.csv or.xlsx files and post-processed identically. Missing abundance values were imputed using the minimum value of each metabolite divided by 2. All abundance values in the matrix were then $\log_2$ transformed and features (metabolites) were auto-scaled by subtracting the mean and dividing by the standard deviation.

**Metabolite identifier harmonisation.**   In order to map compounds to the three pathway databases investigated in this study (KEGG, Reactome, and BioCyc), metabolite identifiers in each dataset were converted to the corresponding identifier type. For the conversion of compound names to KEGG identifiers, the MetaboAnalyst 4.0 [47] ID conversion tool was used (https://www.metaboanalyst.ca/MetaboAnalyst/upload/ConvertView.xhtml). For Reactome, KEGG compounds were mapped to ChEBI identifiers using the Python bioservices package (v 1.7.1) [48]. For BioCyc, the web-based metabolite translation service (https://metacyc.org/metabolite-translation-service.shtml) was used to convert from KEGG to BioCyc identifiers.

**Selection of differentially abundant metabolites.**   The list of metabolites of interest was determined using a series of two-tailed student's t-tests to determine whether each metabolite in the dataset was significantly associated with the outcome of interest. P-values were adjusted using the Benjamini-Hochberg False discovery rate (BH FDR) procedure [49] to account for multiple testing. Significantly differentially abundant (DA) metabolites were then selected based on a q-value threshold of $q \leq 0.05$. To investigate the effect of the list of input metabolites on the number of significant pathways, we used both BH FDR and Bonferroni methods for p-value adjustment and tested several cut-off thresholds (adjusted $p \leq 0.005$, 0.05, or 0.1) for the selection of DA metabolites using each method.

### Performing pathway enrichment

**Pathway database details.** For the purposes of this paper, the pathway sets used contained only compounds (including small molecules, metabolites, and drugs). KEGG pathways and their corresponding compounds were downloaded using the KEGG REST API (https://www.kegg.jp/kegg/rest/keggapi.html) in October 2020, corresponding to KEGG release 96. Reactome pathways release 75 were downloaded from https://reactome.org/download-data. BioCyc pathways v24.5 were exported from https://biocyc.org/ using the SmartTables function.

**ORA implementation.** ORA was implemented using a custom Python script that utilised the scipy stats fisher_exact function (right-tailed) to calculate pathway p-values. Only pathways containing at least 3 compounds were used as input for ORA. p-values were calculated if the parameter k (number of differentially abundant metabolites in the i[th] pathway) was $\geq$ 1.

### Metabolite misidentification

**Implementation details.** All simulations were performed using Python (v 3.8). Simulations with an element of randomisation were repeated 100 times, and results are reported as the mean of 100 random samplings of the simulation, alongside the standard error of the mean.

**Simulating metabolite misidentification.** Chemical formula and molecular weight information for each metabolite was obtained using the KEGG REST API. For each level of metabolite misidentification, we randomly selected $f$% ($f$ = 0, 1, . . .X%) of compounds that had at least one other compound with a molecular weight within ±20ppm (approximately isobaric compound) present in the KEGG pathway set. For each randomly selected compound, one of its isobaric compounds was randomly selected and the identifier of this compound then replaced the original identifier in the dataset, thereby simulating misidentification by mass. Similarly, for misidentification by chemical formula, compounds that had at least one other compound with an identical chemical formula present in the KEGG pathway set were randomly selected, and compound identifiers replaced. Replacement compounds must be present in at least one KEGG pathway but must not already form part of the original background list, to avoid introducing duplicate compounds.

**Quantifying changes in results.** To illustrate how lists of significant pathways change at varying levels of metabolite misidentification, we define two performance statistics: the pathway loss rate and the pathway gain rate. The pathway loss rate represents the proportion of the original pathways (0% misidentification) significant at p $\leq$ 0.1 that are no longer significant at $f$% misidentification. The pathway gain rate represents the proportion of pathways that were not significant at 0% misidentification but become significant at $f$% misidentification.

Let A and B be sets of pathways from ORA such that:

$$A = \{\text{Pathways significant at 0 \% metabolite misidentification (p} \leq 0.1)\}$$

$$B_f = \{\text{Pathways significant at } f \text{ \% metabolite misidentification (p} \leq 0.1)\}$$

The *pathway loss rate* and *pathway gain rate* at $f$% metabolite misidentification are then defined as:

$$Pathway\ loss\ rate\Big(A, B_f\Big) = 1 - \frac{|A \cap B_f|}{|A|} \tag{2}$$

$$Pathway\ gain\ rate\Big(A, B_f\Big) = \frac{|B_f - A|}{|A|} \tag{3}$$

where |A| indicates the cardinality (number of elements) in the set A, and |B-A| indicates the set formed by those members of B which are not members of A.

## Overlap coefficient

To quantify the similarity between pathways, represented by lists of metabolites, we use the overlap coefficient. The overlap (Szymkiewicz–Simpson) coefficient is defined as the size of the intersection of two sets A and B, divided by the size of the smallest set.

$$OC(A, B) = \frac{|A \cap B|}{min(|A|, |B|)}.$$

## Supporting information

**S1 Supporting Information. Supplementary figures and tables. Fig A: The effect of the number of input metabolites on the number of significant pathways (p $\leq$ 0.1) across all datasets.** All metabolites in the dataset were ranked by their raw p-value which was calculated using t-tests to determine the level of differential abundance between two study groups. Beginning with the compound with the lowest p-value, the list of DA metabolites was created by adding one compound at a time (x-axis). ORA was performed using this list and the number of significant pathways at p $\leq$ 0.1 is shown on the y-axis. Bonferroni adjusted p-value thresholds are indicated using red markers and BH-FDR adjusted q-values are indicated using black markers. **Table A: Significant pathways (P $\leq$ 0.1) obtained with KEGG, HumanCyc, and Reactome using the Yachida et al. dataset.** Pathways with similar biological function significant at P $\leq$ 0.1 using at least two pathway databases are highlighted in bold. **Fig B: Overlap coefficient values between all metabolites in significant pathways (p $\leq$ 0.1) detected using KEGG and Reactome. Fig C: Metabolite misidentification.** Heatmaps showing pathway loss rate and pathway gain rate at varying percentages of metabolite misidentification by (a) identical chemical formula and (b) molecular mass within a +/- 20ppm window. Colour bar corresponds to pathway loss/gain rate, with darker colours representing lower rates. Misidentification by chemical formula shown up to 5%, whereas misidentification by mass shown up to 6%, as these are the highest values calculatable (based on limited replacement compounds) across all datasets.
(DOCX)

## Acknowledgments

The authors gratefully acknowledge the help of the Reactome support team based at the Ontario Institute for Cancer Research, for providing previous release files of their database.

## Author Contributions

**Conceptualization:** Cecilia Wieder, Clément Frainay, Jacob G. Bundy, Fabien Jourdan, Timothy Ebbels.

**Data curation:** Florence Vinson.

**Formal analysis:** Cecilia Wieder.

**Investigation:** Cecilia Wieder.

**Methodology:** Cecilia Wieder, Clément Frainay, Fabien Jourdan, Timothy Ebbels.

**Software:** Cecilia Wieder.

**Supervision:** Rachel PJ Lai, Fabien Jourdan, Timothy Ebbels.

**Validation:** Pablo Rodríguez-Mier, Juliette Cooke.

**Visualization:** Cecilia Wieder.

**Writing – original draft:** Cecilia Wieder.

**Writing – review & editing:** Cecilia Wieder, Clément Frainay, Nathalie Poupin, Pablo Rodríguez-Mier, Juliette Cooke, Rachel PJ Lai, Jacob G. Bundy, Fabien Jourdan, Timothy Ebbels.

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
