## [Decision Letter · Decision Letter 0]

23 Jun 2021

Dear Dr Ebbels,

Thank you very much for submitting your manuscript "Pathway analysis in metabolomics: pitfalls and best practice for the use of over-representation analysis" for consideration at PLOS Computational Biology.

As with all papers reviewed by the journal, your manuscript was reviewed by members of the editorial board and by several independent reviewers. In light of the reviews (below this email), we would like to invite the resubmission of a significantly-revised version that takes into account the reviewers' comments.

We cannot make any decision about publication until we have seen the revised manuscript and your response to the reviewers' comments. Your revised manuscript is also likely to be sent to reviewers for further evaluation.

Sincerely,

Kiran Raosaheb Patil, Ph.D.

Deputy Editor

PLOS Computational Biology

Jason Papin

Editor-in-Chief

PLOS Computational Biology

Reviewer's Responses to Questions

**Comments to the Authors:**

Reviewer #1: Wieder et al examine how different parameters for pathway over-representation analysis (ORA) influence the results of metabolomics data analysis. They use five experimental metabolomics data sets (from humans, mouse and E. coli) to test the relationship between ORA parameters and ORA results. The study is relevant for the field because it nicely illustrates the strong influence of ORA parameters on the outcome of the analysis. However, my main concern is that the authors cannot identify the best parameter configuration because they lack a proper reference of what is true (they mention this also in the discussion). Specific comments are below:

1) The authors claim in the abstract that they used in-silico simulations, thus I expected that they simulated metabolic changes (e.g. with a dynamic model) and used ORA to recover the true in silico perturbation. Instead they use real data, which is great, but it is difficult to judge which parameter configuration is best. The authors mention themselves that the study lacks a “ground-truth dataset”. The authors should at least better describe the nature of such a ground-truth dataset/ result. They should also describe better what they mean with in silico simulation.

2) The authors selected 5 experimental data sets for their study. They could better describe in the main text (instead of Table 1) why they selected these data and which conditions/organisms are investigated. For example, why did they select (only) two strains out of 3800 E. coli strains in Fuhrer et al? In fact, these data contain some information about the "ground truth", because in many cases the deleted gene can be assigned to a metabolic pathway.

3) A main concern is the selection of Databases. Obviously, the best choice is a genome-scale reconstruction of metabolism of the respective organism and I wonder why the authors did not consider them; at least for the E. coli data, mouse and HeLa cells.

4) The authors could give a better overview about the parameters tested and better quantify their relevance relative to each other. The recommendations in the discussion are not specific enough. For example, how could one derive a “consensus” pathway signature.

Reviewer #2: The authors assess parameters used in pathway enrichment analysis using 5 publicly available MS-based metabolomics datasets. While those dealing with these tools have surely identified inconsistencies in results according to the tools and parameters used, the exercise of testing the boundaries and consequences in results of mis-use of the tools is interestingly quantified by the authors. In addition it is of value the section on recommendations on the best practice to use over-representation analysis (ORA) in the metabolomics field.

The manuscript is well-written but requires improvement in certain sections.

Title/introduction – It is worth mentioning that ORA is also known as metabolite enrichment analysis, that might even be a more common name used within the metabolomics community.

Methods

L501 – for dataset MTBLS135 the text mentions that the sample type is plasma, while Table 1 mentions tissue. So here it is important to rectify and harmonize. In addition, the files of the uploaded dataset mention ‘serum’ and not plasma. This might sound like a detail, but one should be precise, as the two sample types (serum and plasma) are not interchangeable.

L502 – dataset MTBLS136: I could not retrieve any data files in the Metabolights repository for this study! Supplementary Materials of the associated publication do not contain the metabolomics data itself per sample. So I could also not confirm the number of samples (controls and estrogen-users).

L506 – as for all the other datasets, it is important to mention the number of samples for the last dataset Fuhrer et al. And was the negative mode subdataset used or the positive mode or both? In the results, it is then mentioned 2 subsets from this particular study, so this needs to be clarified in the Methods.

Table 1 – where does the total number of metabolites mapping to KEGG compounds was extracted from? Analysis within this manuscript or extracted from the original datasets?

L525 – metabolite ID conversion

This is a stress point of identification and according to the algorithms used, it can over-identify and thus overestimate metabolite coverage or if too conservative, it can assign only a part of possible metabolites.

For example: when one measures an amino acid, will it be immediately assigned to L-amino acid? An amino acid can be also D-amino acid in a biological environment, however this type of assignment is hardly assessed and possibly not even feasible to know using regular LC/CE-MS techniques (one would need to use chiral chromatography fo example). And in the likely even of not knowing, will it be assigned to D/L-amino acid or assumed to be L-amino acid?

Another example are acids and salts and ions (for example: glutamic acid vs glutamate vs sodium glutamate (or any other salt)): will these be assigned to the same metabolite ID or to different ones?

As different metabolite ID convertors (tool in MetaboAnalyst, too in BioCyc, etc) were used, it is likely that these will produce different results!! This aspect deserves some explanation and words of caution in the manuscript. Will the IDs be back-converted to the same list of IDs when using convertors from other databases? This would be good to check.

L567 – metabolite misidentification

Results

L169 – NMR is not relevant in this study, as none of the studies chosen have used it, so please remove it.

L293 – if the authors want to mention MSI levels 2-4, then they need to explain what these are, as the readers might not know…

Fig5 – A and B figures are actually quite similar. So it might be worth mentioning in the text that the misidentification is probably from molecular formula to metabolite and not so much from mass to molecular formula. Some words on the similarly / differences between these two graphs are worth mentioning.

L334 - 349 – one needs to be careful with these type of statements. Reversed phase is used in combination with ion pairing for detecting polar metabolites, of a similar nature to the ones that are detected by HILIC. HILIC can also detect a lot of apolar metabolites, because it can act in a mixed mode type of chromatography. In addition GC-MS with a prior derivatisation step in the sample preparation has been used a lot for detecting polar metabolites! So being that there is a lot of variety in analytical and sample prep methods for metabolomics, this whole section should be rephrased and adapted.

The authors should stick to polarity of compounds to make their point, irrespective of the technique used, as clearly the reality is not this simple, as it does not only depend on chromatography!! in fact one of the datasets does not use chromatography but capillary electrophoresis!!

Then none of the datasets aimed at lipid metabolism, this would then lead to completely different result. So this whole section is very circumstantial and simply not informative.

Discussion

L395 – mis-identification is abundant in all analytical platforms!

L397 – not relevant to mention NMR as it was not used in this study. To add to this: maybe NMR provides less coverage but maybe better identification…?

Reviewer #3: Wieder and colleagues performed an interesting study on the application of ORA to metabolomics data. The paper is well-written and proposes, for the first time, the guidelines to perform ORA analysis in metabolomics. I especially enjoyed reading the pathway comparison part, it is a nice addition to the paper. However, some of the observations or conclusions were somewhat trivial to me. Still, I find the paper suitable for publication and I suggest the following changes to improve the paper:

- The authors state: "To perform ORA, three essential inputs are required: a collection of pathways (or custom metabolite sets), a list of metabolites of interest, and a background or reference set." By definition, all annotatable metabolites in untargeted metabolomics are all those in the collection of pathways. How do all annotatable metabolites and all metabolites in the pathway differ?

- Pg 12, section "increasing the number...". It is not needed to do all that to demonstrate this trivial aspect. It is expected. The authors could perform a similar approach but instead of considering all pathways, considering only those pathways that have at least 2 (or 3 if data allows it) DA, and then randomly add new DA to the pathways to see how the overall ranking fluctuates. Otherwise, adding DA by p-value is arbitrary and, considering the nature of untargeted metabolomics data, these observations are expected.

- "Pathway sets can be obtained freely from several databases..". . KEGG is partially commercial so should not be included. For BioCyc, I would like to know how the authors obtained that information as I believe it's partially commercial. MetExplore uses others databases so it should be removed as well. Is Ingenuity still on business?

- Pg 26: "Suggested recommendations...". The paper discusses the ambiguity of the composition of the background set in untargeted metabolomics, but the recommendations are not clear on how this background set should be built in untargeted metabolomics. It would be worthwhile to break down the first recommendation into untargeted and targeted.

- Discussion, how using topology-based or FCS do/could naturally overcome some of the ORA limitations, or introduce different biases. Could the recommendations be instead: do not use ORA, but FCS/Topology-based? What could the limitations of FCS/Topology-based in untargeted metabolomics be? I believe a brief discussion about this is necessary.

- Pg 7 lines 139; "consisting of all compounds annotated to at least one KEGG pathway", could you define this better?

Minor:

- Change: Firstly -> first, secondly -> second

- Pg 5 line 95: p-value, P should be capitalized.

- Pg 14 line 225. "Pathway database is key" I suggest using a more informative sentence.

- I did not find the supplementary materials.

**Have the authors made all data and (if applicable) computational code underlying the findings in their manuscript fully available?**

Reviewer #1: Yes

Reviewer #2: Yes

Reviewer #3: Yes

PLOS authors have the option to publish the peer review history of their article (what does this mean?). If published, this will include your full peer review and any attached files.

Reviewer #1: No

Reviewer #2: **Yes: **Sofia Moco

Reviewer #3: No
---

## [Decision Letter · Decision Letter 1]

12 Aug 2021

Dear Dr Ebbels,

Thank you very much for submitting your manuscript "Pathway analysis in metabolomics: pitfalls and best practice for the use of over-representation analysis" for consideration at PLOS Computational Biology. As with all papers reviewed by the journal, your manuscript was reviewed by members of the editorial board and by several independent reviewers. The reviewers appreciated the attention to an important topic. Based on the reviews, we are likely to accept this manuscript for publication, providing that you modify the manuscript according to the editorial recommendation below.

Before formal acceptance, I would like to suggest a change in the title: replacing "pitfalls and best practice" by "recommendations". The reason being that the term "best" in the computational context often implies optimisation / rigorous analytical basis. I would therefore like to encourage you to consider this change (and consistent changes in the rest of the manuscript along these lines).

Sincerely,

Kiran Raosaheb Patil, Ph.D.

Deputy Editor

PLOS Computational Biology

Jason Papin

Editor-in-Chief

PLOS Computational Biology

[LINK]

As summarised below, the reviewers are satisfied with the response and the changes to the manuscript. Before formal acceptance, I would like to suggest a change in the title: replacing "pitfalls and best practice" by "recommendations". The reason being that the term "best" in the computational context often implies optimisation / rigorous analytical basis. I would therefore like to encourage you to consider this change (and consistent changes in the rest of the manuscript along these lines).

Reviewer's Responses to Questions

**Comments to the Authors:**

Reviewer #1: I thank the authors for addressing all of my points, and I have no further comments.

Reviewer #2: The authors improved the study by addressing the reviewers concerns to a level that in my opinion makes this manuscript worthy of publication.

Reviewer #3: The authors have addressed all my concerns.

**Have the authors made all data and (if applicable) computational code underlying the findings in their manuscript fully available?**

Reviewer #1: None

Reviewer #2: None

Reviewer #3: None

PLOS authors have the option to publish the peer review history of their article (what does this mean?). If published, this will include your full peer review and any attached files.

Reviewer #1: No

Reviewer #2: **Yes: **Sofia Moco

Reviewer #3: No

Figure Files:

Data Requirements:

Reproducibility:

References:

---

## [Editor Report · Decision Letter 2]

23 Aug 2021

Dear Dr Ebbels,

We are pleased to inform you that your manuscript 'Pathway analysis in metabolomics: recommendations for the use of over-representation analysis' has been provisionally accepted for publication in PLOS Computational Biology.

Best regards,

Kiran Raosaheb Patil, Ph.D.

Deputy Editor

PLOS Computational Biology

Jason Papin

Editor-in-Chief

PLOS Computational Biology

---

## [Editor Report · Acceptance letter]

2 Sep 2021

PCOMPBIOL-D-21-00895R2 

Pathway analysis in metabolomics: recommendations for the use of over-representation analysis

Dear Dr Ebbels,

I am pleased to inform you that your manuscript has been formally accepted for publication in PLOS Computational Biology. Your manuscript is now with our production department and you will be notified of the publication date in due course.

With kind regards,

Katalin Szabo
